# Hepatitis B virus-related hepatocellular carcinoma has superior overall survival compared with other etiologies

Yi-Hao Yen[1]☯*, Kwong-Ming Kee[1], Tsung-Hui Hu[1], Ming-Chao Tsai[1], Yuan-Hung Kuo[1], Wei-Feng Li[2], Yueh-Wei Liu[2], Chih-Chi Wang[2]☯*, Chih-Yun Lin[3]

1 Division of Hepatogastroenterology, Department of Internal Medicine, Kaohsiung Chang Gung Memorial Hospital and Chang Gung University College of Medicine, Kaohsiung, Taiwan, 2 Liver Transplantation Center and Department of Surgery, Kaohsiung Chang Gung Memorial Hospital, Kaohsiung, Taiwan, 3 Biostatistics Center of Kaohsiung Chang Gung Memorial Hospital, Kaohsiung, Taiwan

☯ These authors contributed equally to this work.
* cassellyen@yahoo.com.tw (YHY); ufel4996@ms26.hinet.net (CCW)

## Abstract

### Background

Whether the etiology of chronic liver disease (CLD) impacts the overall survival (OS) of patients with hepatocellular carcinoma (HCC) remains unclear. We aim to clarify this issue.

### Materials and methods

Between 2011 and 2020, 3941 patients who were newly diagnosed with HCC at our institution were enrolled in this study. In patients with multiple CLD etiologies, etiology was classified using the following hierarchy: hepatitis C virus (HCV) > hepatitis B virus (HBV) > alcohol-related > all negative. All negative was defined as negative for HCV, HBV, and alcohol use disorder.

### Results

Among 3941 patients, 1407 patients were classified with HCV-related HCC, 1677 patients had HBV-related HCC, 145 patients had alcohol-related HCC, and 712 patients had all-negative HCC. Using the all-negative group as the reference group, multivariate analysis showed that HBV is an independent predictor of mortality (hazard ratio: 0.856; 95% confidence interval: 0.745–0.983; p = 0.027). Patients with HBV-related HCC had superior OS compared with patients with other CLD etiologies (p<0.001). Subgroup analyses were performed, for Barcelona Clinic Liver Cancer (BCLC) stages 0–A (p<0.001); serum alpha-fetoprotein (AFP) levels≧20 ng/ml (p<0.001); AFP levels < 20 ng/ml (p<0.001); age > 65 years (p<0.001); and the use of curative treatments (p = 0.002). No significant difference in OS between HBV and other etiologies was observed among patients aged ≤ 65 years (p = 0.304); with BCLC stages B–D (p = 0.973); or who underwent non-curative treatments (p = 0.1).

**Editor:** Alessandro Granito, University Hospital of Bologna Sant'Orsola-Malpighi Polyclinic Department of Digestive System: Azienda Ospedaliero-Universitaria di Bologna Policlinico Sant'Orsola-Malpighi Dipartimento dell'apparato digerente, ITALY

**Data Availability Statement:** Raw data for the cohort involved in this study is available via the following digital object identifier: https: https://www.dropbox.com/scl/fi/ijxmur8uh3clb5lo0i3ys/n-3941.xlsx?rlkey=wooxyhuaops6pp5ybk31am1gp&dl=0.

**Funding:** • Initials of the authors who received each award: YHY • Grant numbers awarded to each author: CMRPG8N1131 • The full name of each funder: Kaohsiung Chang Gung Memorial Hospital • URL of each funder website: https://cghdpt.cgmh.org.tw/branch/shk • the funder had no role in the study design, data collection and analysis, decision to publish, or preparation of the manuscript.

**Competing interests:** The authors have declared that no competing interests exist.

## Conclusion

Patients with HBV-related HCC had superior OS than patients with other HCC etiologies.

## Introduction

Hepatocellular carcinoma (HCC) is the most common primary hepatic malignant tumour and represents an important global health problem. It is currently the third leading cause of cancer-related deaths in the general population and the first among cirrhotic patients. Moreover, its incidence has continuously increased, with the number of affected cases more than tripling since 1980 [1].

HCC is a highly heterogeneous cancer at both the genetic and histological levels [2]. A recent study found higher mutation rates in the gene encoding activin A receptor type 2A among patients with nonalcoholic steatohepatitis (NASH)-associated HCC than among patients with HCC due to other etiologies (10% vs. 3%, $p<0.05$). In vitro studies indicate that activin A receptor type 2A functions as a tumor suppressor [3]. These findings suggest that tumor biology may vary according to the etiology of chronic liver dis-ease (CLD).

About one-third of HCC can be classified into specific histological subtypes representing discrete entities with prognostic implications [4]. Steatohepatitic HCC is a HCC subtype that frequently occurs in patients with NASH [5]. Its molecular appearance is similar to normal liver, lacking Want/beta-catenin pathway activation and leading to a relatively good prognosis. On MRI, steatohepatitis HCC typically shows intralesional fat content [4] Macro trabecular-massive HCC frequently develops in patients with hepatitis B virus (HBV) infection and is associated with a poor prognosis and an aggressive phenotype, including high serum alpha-fetoprotein (AFP) levels and pathologic features such as satellite nodules and vascular invasion [6]. The main radiological feature of this form is an arterial phase hypo vascular component [7].

A multicenter study evaluated HCC growth patterns by quantifying the tumor doubling time (TDT). Indolent tumor growth was defined as a TDT >365 days, and rapid tumor growth was defined as a TDT <90 days. Indolent growth was more common in non-viral than viral cirrhosis (50.9% vs. 32.1%) [8]. A systemic review and meta-analysis demonstrated that a HBV etiology and poor tumor differentiation were associated with rapid tumor growth in HCC [9].

Based on the results of these studies [2–9], the tumor biology of HCC may vary ac-cording to the CLD etiology. We hypothesize that the CLD etiology impacts overall survival (OS) among patients with HCC.

The advent of highly effective antiviral therapies for HBV and hepatitis C virus (HCV) [10, 11], combined with the obesity epidemic, has resulted in a decreasing prevalence of virus-related HCC, and nonalcoholic fatty liver disease (NAFLD) is the most rapidly growing HCC etiology in developed countries. Younossi et al. examined the HCC cases available in the Surveillance, Epidemiology and End Results (SEER) registries of Medicare-associated data. A total of 4,929 HCC cases were enrolled between 2004–2009, including 3207 patients with virus-related HCC (65%) and 701 patients with NAFLD-related HCC (14.1%) [12]. Karim et al. enrolled a cohort of HCC patients from the SEER Medicare-associated registries between 2011 and 2015. Among 5098 HCC patients, NAFLD was the leading etiology, accounting for 1813 cases (35.6%), followed by virus-related HCC, accounting for 1961 cases (38.4%) [13]. However, contemporary large-scale studies evaluating the impacts of CLD etiology on OS among patients with HCC are limited. We evaluated whether the etiology of CLD impacts OS among an East Asian cohort of patients with HCC.

## Methods

Data were extracted from Kaohsiung Chang Gung Memorial Hospital's HCC registry database of prospectively collected and annually updated data. Between August 8, 2022 and August 7, 2023, the data were accessed for research purposes.

The authors had no access to information that could identify individual participants during or after data collection.

From 2011 to 2020, 3977 patients were newly diagnosed with HCC at our institution. The exclusion criteria are as follows:

Age<18 years (n = 7).

Hepatitis B surface antigen (HBsAg) or anti-HCV unknown (n = 1).

Unknown alcohol intake history among patients negative for both HBsAg and anti-HCV (n = 28). The remaining 3941 patients were enrolled in this study.

### Variables of interest

Patient information, including height, weight, age, and sex, was retrieved from the database. Tumor size was determined according to either pathological examinations in patients who underwent surgery or imaging findings in patients who underwent non-surgical treatments. Tumor number (solitary vs. multiple) was determined based on imaging results. The presence of cirrhosis was indicated by either an Ishak score [14] of 5 or 6 in patients who underwent surgery or imaging results in patients who underwent non-surgical treatments. Cirrhosis was indicated if imaging results showed evidence of cirrhosis, such as small liver size, nodular liver surface, or the presence of regeneration nodules [15]. HCC is diagnosed either through imaging studies or histopathological examinations. The imaging diagnoses of HCCs were made through computed tomography (CT) or magnetic resonance imaging (MRI) in cirrhotic patients with nodule(s) ≥1 cm, arterial phase hyperenhancement, and washout in the portal venous or delayed phases according to the recommendations of the European Association for the Study of the Liver (EASL) guidelines [16]. Barcelona Clinic Liver Cancer (BCLC) stages [17] were defined according to the original definitions, and BCLC stage A was defined using the Milan criteria [18]. Other data, including imaging-diagnosed tumor–node–metastasis (TNM) stage (based on the 7th edition of the American Joint Committee on Cancer [AJCC]) [19], serum AFP level, Child–Pugh class [20], the presence of HBsAg, the presence of anti-HCV antibody, in-Ter national normalized ratio (INR), creatinine level, bilirubin level, alcohol intake history, and HCC diagnostic method (i.e., imaging vs. pathological diagnosis), were prospectively collected from the HCC registry.

Patients with HCV infections were defined based on anti-HCV antibody positivity. Patients with HBV infections were defined based on HBsAg positivity. Patients were classified as having an alcohol-related etiology if they reported frequent alcohol intake. Patients were classified as all negative if they were negative for both anti-HCV antibody and HBsAg and did not have a history of frequent alcohol intake. For patients with multiple etiologies, classification was performed using the following hierarchy: HCV > HBV > alcohol-related > all negative [21].

Curative treatments were defined as liver transplantation, liver resection, or ablation. Non-curative treatments included transcatheter arterial embolization (TAE)/transcatheter arterial chemoembolization (TACE), targeted therapy (e.g., sorafenib or Lenvatinib), systemic chemotherapy, radiation therapy, and best supportive care.

### Antiviral therapies

At our institution, antiviral therapies with nucleos(t)ide analogs are administered to treat chronic HBV infections according to the following guidelines: (1) patients present with

**Table 1. Characteristics of patients according to etiologies of chronic liver disease.**

|  | HCV, n = 1407 | HBV, n = 1677 | Alcohol use, n = 145 | All negative, n = 712 | p |
|---|---|---|---|---|---|
| Age (years) | 66 (60–73) | 60 (52–67) | 59(51.5–65) | 68(60–76) | <0.001 |
| Male | 829 (58.9%) | 1381(82.3%) | 141(97.2%) | 492(69.1%) | <0.001 |
| BMI (kg/m$^2$) | 24.5(22.3–27.2) | 24.5(22.1–27.2) | 24.8(22.0–27.5) | 25.0(22.7–28.0) | 0.032 |
| Cirrhosis |  |  |  |  | <0.001 |
| Presence | 1055(75.0%) | 1146(68.3%) | 100(69.0%) | 439(61.7%) |  |
| Absence | 348(24.7%) | 530(31.6%) | 44(30.3%) | 267(37.5%) |  |
| Unknown | 4(0.3%) | 1(0.1%) | 1(0.7%) | 6(0.8%) |  |
| Creatinine (mg/dl) | 1.0(0.8–1.3) | 1.0(0.8–1.2) | 1.1(0.9–1.4) | 1.1(0.8–1.5) | <0.001 |
| Total bilirubin (mg/dl) | 1.1(0.8–1.6) | 1.0(0.8–1.6) | 0.8(1.3–2.3) | 1.0(0.7–1.5) | <0.001 |
| INR | 1.0(1.0–1.1) | 1.0(1.0–1.1) | 1.1(1.0–1.2) | 1.0(1.0–1.1) | <0.001 |
| Child Pugh class |  |  |  |  | <0.001 |
| A | 1120(79.6%) | 1387(82.7%) | 100(69.0%) | 583(81.9%) |  |
| B | 229(16.3%) | 210(12.5%) | 39(26.9%) | 98(13.8%) |  |
| C | 38(2.7%) | 62(3.7%) | 6(4.1%) | 14(2.0%) |  |
| Unknown | 20(1.4%) | 18(1.1%) | 0 | 17(2.4%) |  |
| Method of HCC diagnosis |  |  |  |  | 0.001 |
| Image | 551(39.2%) | 624(37.2%) | 68(46.9%) | 227(31.9%) |  |
| Pathological | 856(60.8%) | 1053(62.8%) | 77(53.1%) | 485(68.1%) |  |
| Tumor size (mm) | 30(21–50) | 35(23–74) | 32(21.5–82.5) | 47.5(28–93) | <0.001 |
| Tumor number by imaging studies |  |  |  |  | 0.446 |
| Single | 864(61.4%) | 1023(61.0%) | 79(54.5%) | 434(61.0%) |  |
| Multiple | 543(38.6%) | 654(39.0%) | 66(45.5%) | 278(39.0%) |  |
| 7$^{th}$ edition AJCC stage |  |  |  |  | <0.001 |
| 1 | 739(52.5%) | 818(48.8%) | 60(41.4%) | 234(45.5%) |  |
| 2 | 315(22.4%) | 282(16.8%) | 39(26.9%) | 115(16.2%) |  |
| 3 | 248(17.6%) | 388(23.1%) | 26(17.9%) | 178(25.0%) |  |
| 4 | 88(6.3%) | 161(9.6%) | 17(11.7%) | 79(11.1%) |  |
| Unknown | 17(1.2%) | 28(1.7%) | 3(2.1%) | 16(2.2%) |  |
| BCLC stage |  |  |  |  | <0.001 |
| 0 | 212(15.3%) | 217(13.2%) | 18(12.7%) | 47(6.8%) |  |
| A | 581(42.0%) | 589(35.8%) | 45(31.7%) | 205(29.7%) |  |
| B | 239(17.3%) | 342(20.8%) | 34(23.9%) | 186(26.9%) |  |
| C | 281(20.3%) | 415(20.2%) | 36(25.4%) | 210(30.4%) |  |
| D | 69(5.0%) | 82(5.0%) | 9(6.3%) | 43(6.2%) |  |
| AFP |  |  |  |  | 0.001 |
| ≧20 ng/ml | 738(52.5%) | 891(53.1%) | 64(44.1%) | 322(45.2%) |  |
| <20 ng/ml | 669(47.5%) | 786(46.9%) | 81(55.9%) | 390(54.8%) |  |
| Treatment |  |  |  |  | <0.001 |
| Transplant | 54(3.8%) | 58(3.5%) | 5(3.4%) | 16(2.2%) |  |
| Resection | 398(28.3%) | 635(37.9%) | 38(26.2%) | 286(33.1%) |  |
| Ablation | 411(29.2%) | 314(18.7%) | 33(22.8%) | 133(18.7%) |  |
| Best supportive care | 73(5.2%) | 84(5.0%) | 12(8.3%) | 41(5.8%) |  |
| Chemotherapy | 10(0.7%) | 35(2.1%) | 2(1.4%) | 14(2.0%) |  |
| TAE/TACE | 326(23.2%) | 318(19.0%) | 39(26.9%) | 166(23.3%) |  |
| Target therapy | 94(6.7%) | 183(10.9%) | 11(7.6%) | 75(10.5%) |  |

(*Continued*)

**Table 1.** (Continued)

|  | HCV, n = 1407 | HBV, n = 1677 | Alcohol use, n = 145 | All negative, n = 712 | p |
|---|---|---|---|---|---|
| Radiation therapy | 41(2.9%) | 50(3.0%) | 5(3.4%) | 31(4.4%) | |

AFP, alpha-fetopotein; BMI, body mass index; HCV, hepatitis C virus; HBV, hepatitis B virus; INR, international normalized ratio; AJCC, American Joint Committee on Cancer; BCLC, Barcelona clinic liver cancer; TACE, Transcatheter Arterial Chemoembolization; TAE, Transcatheter Arterial embolization

persistently elevated alanine transaminase (ALT) and HBV DNA; (2) cirrhotic patients or patients with HCC who underwent curative treatments and present with elevated HBV DNA, irrespective of ALT levels; or (3) patients with liver decompensation [10]. Antiviral therapies for chronic HCV infection consisted of pegylated interferon combined with ribavirin from 2011 to 2016; this therapy was re-placed with direct-acting antiviral agents (DAA) starting in January 2017. Anti-HCV therapies were indicated for patients who were anti-HCV antibody positive with detectable serum HCV RNA [11].

## Statistical analysis

Variables are presented as the number and percentage or the median and interquar-tile range. The Chi-square test was used to compare categorical variables. The Kruskal–Wallis test was used to compare continuous variables. OS was defined as the time from the date of HCC diag-nosis to the date of death or last follow-up. Comparisons of OS between groups were per-formed using Kaplan–Meier survival analyses and the log-rank test. Univariable and multivariable Cox proportional hazard models were developed to identify variables associated

**Table 2. Univariate and multivariate analysis of variables associated with mortality.**

|  | Univariate | | Multivariate | |
|---|---|---|---|---|
|  | HR (95%CI) | p | HR (95%CI) | p |
| Age (years) ≦65 | Reference | | Reference | |
| >65 | 1.316(1.193–1.451) | <0.001 | 1.141(1.030–1.264) | 0.011 |
| BCLC stage | | | | |
| 0-A | Reference | | Reference | |
| B-D | 4.045(3.629–4.508) | <0.001 | 2.649(2.359–2.974) | <0.001 |
| AFP (ng/ml) | | | | |
| <20 | Reference | | Reference | |
| ≧20 | 2.393(2.161–2.651) | <0.001 | 1.799(1.618–1.999) | <0.001 |
| Treatments | | | | |
| Curative | Reference | | Reference | |
| Non-curative | 4.635(4.177–5.143) | <0.001 | 3.202(2.869–3.575) | <0.001 |
| Etiology | | | | |
| All negative | Reference | | | |
| HCV | 0.765(0.668–0.877) | <0.001 | 0.936(0.814–1.076) | 0.351 |
| HBV | 0.756(0.662–0.864) | <0.001 | 0.856(0.745–0.983) | 0.027 |
| Alcohol use disorder | 1.029(0.791–1.339) | 0.830 | 1.020(0.780–1.334) | 0.885 |

AFP, alpha-fetopotein; HCV, hepatitis C virus; HBV, hepatitis B virus; BCLC, Barcelona clinic liver cancer; Curative treatment was defined as liver transplantation, liver resection, or ablation. Non-curative treatments included transcatheter arterial embolization (TAE)/transcatheter arterial chemoembolization (TACE), target therapy (i.e., sorafenib or lenvatinib), systemic chemotherapy, radiation therapy, and best supportive care.

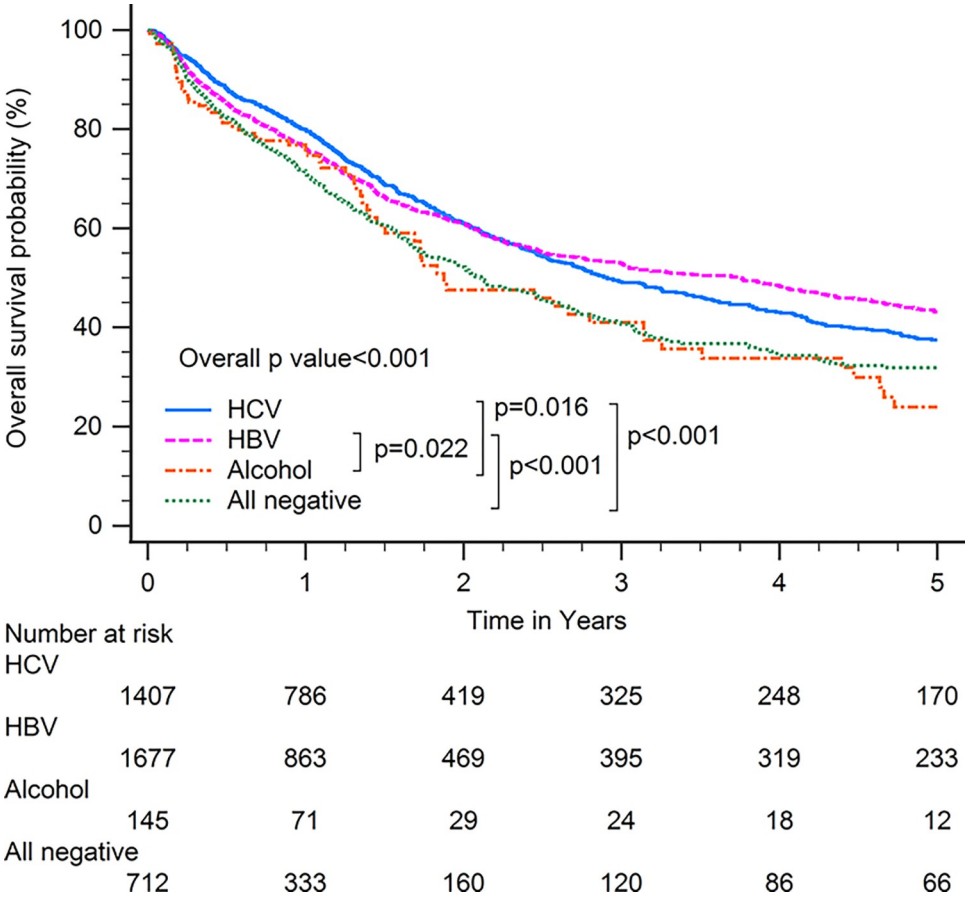

**Fig 1. Overall survival among all patients with hepatocellular carcinoma according to chronic liver disease etiology.**

with mortality. Covariates in the multivariable model were selected *a priori* according to clinical importance. Potential confounders included age, BCLC stage, AFP level, CLD etiology, and treatment received. All potential confounding variables were always retained in multivariable analyses. All statistical analyses were performed using SPSS version 25.0 and MedCalc version 20.110. Two-tailed significance values were applied, with significance defined as $p < 0.05$.

## Results

### Characteristics of patients according to CLD etiology

The all-negative group was older ($p < 0.001$) than the other groups. The proportion of men was smaller in the HCV group ($p < 0.001$) than in the other groups. Body mass index (BMI) was higher in the all-negative group ($p = 0.032$) than in the other groups. The proportion of patients without cirrhosis was larger in the all-negative group ($p < 0.001$) than in the other groups. The creatinine level was higher in the all-negative group ($p < 0.001$) than in the other groups. The bilirubin level was higher in the alcohol-related group ($p < 0.001$) than in the other groups. The INR level was higher in the alcohol-related group ($p < 0.001$) than in the other groups. The proportion of patients with Child–Pugh class A was smaller in the alcohol-related group ($p < 0.001$) than in the other groups. The proportion of patients with HCC diagnosed pathologically was smaller in the alcohol-related group ($p = 0.001$) than in the other groups. The proportion of

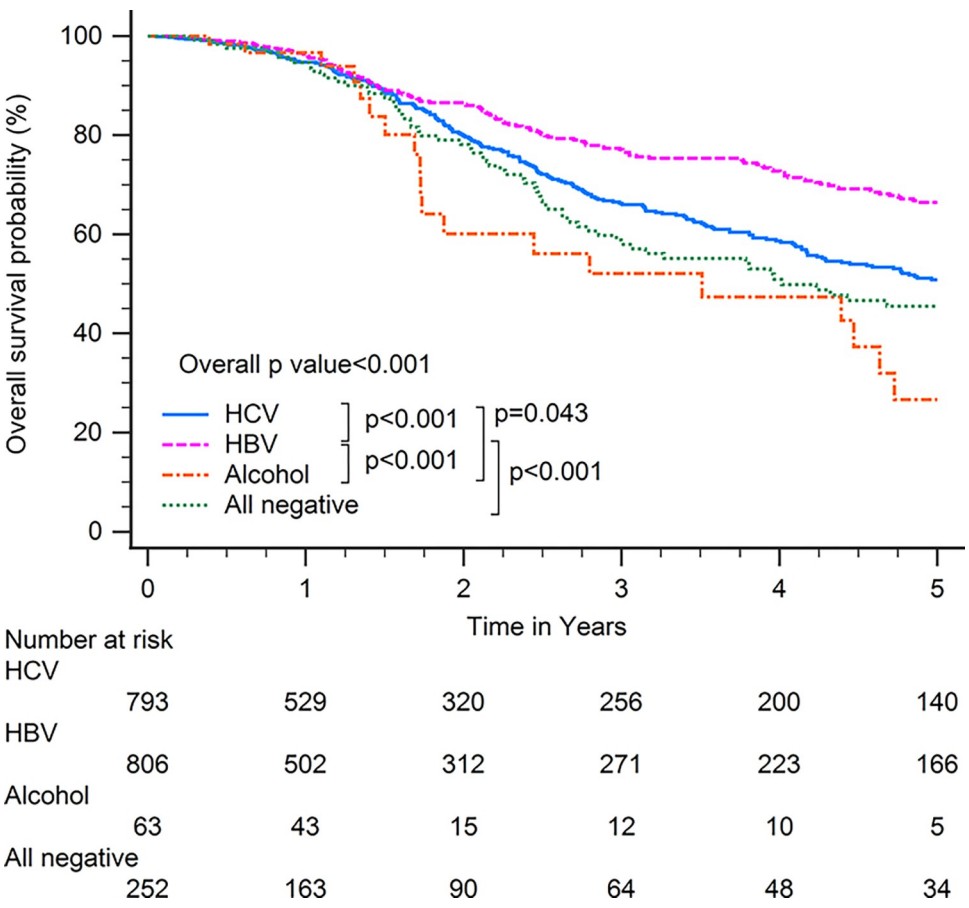

**Fig 2. Overall survival among patients with hepatocellular carcinoma classified as Barcelona clinic liver cancer stages 0–A, according to chronic liver disease etiology.**

patients with AFP ≥20 ng/dl was larger in the HBV group (p = 0.001) than in the other groups. The tumor size was larger in the all-negative group (p<0.001) than in the other groups. However, no significant difference in tumor number was observed among groups. The proportions of patients with BCLC stages 0 and A were larger in the HCV group (p<0.001) than in the other groups. The proportions of patients with TNM stages I and II were larger in the HCV group (p<0.001) than in the other groups. The proportion of patients who underwent resection was larger in the HBV group than in the other groups. The proportion of patients who underwent ablation was larger in the HCV group than in the other groups. The proportion of patients who underwent TAE/TACE was larger in the alcohol-related group than in the other groups (p<0.001) (Table 1).

## Univariate and multivariate analysis of variables associated with mortality

Univariate analysis showed that age >65 years (hazard ratio [HR]: 1.316; 95% con-fidence interval [CI]:1.193–1.451; p<0.001), BCLC stages B–D (HR: 4.045; 95% CI: 3.629–4.508; p<0.001), AFP≧20 ng/ml (HR: 2.393; 95% CI: 2.161–2.651; p<0.001), and non-curative treat-ments (HR: 4.635; 95% CI: 4.177–5.143; p<0.001) were significantly as-sociated with mortal-ity. Using the all-negative group as the reference group, significant differences in mortality were observed for the HCV group (HR: 0.765; 95% CI: 0.668–0.877; p<0.001) and the HBV group (HR: 0.756; 95% CI = 0.662–0.864; p<0.001), whereas no sig-nificant difference was

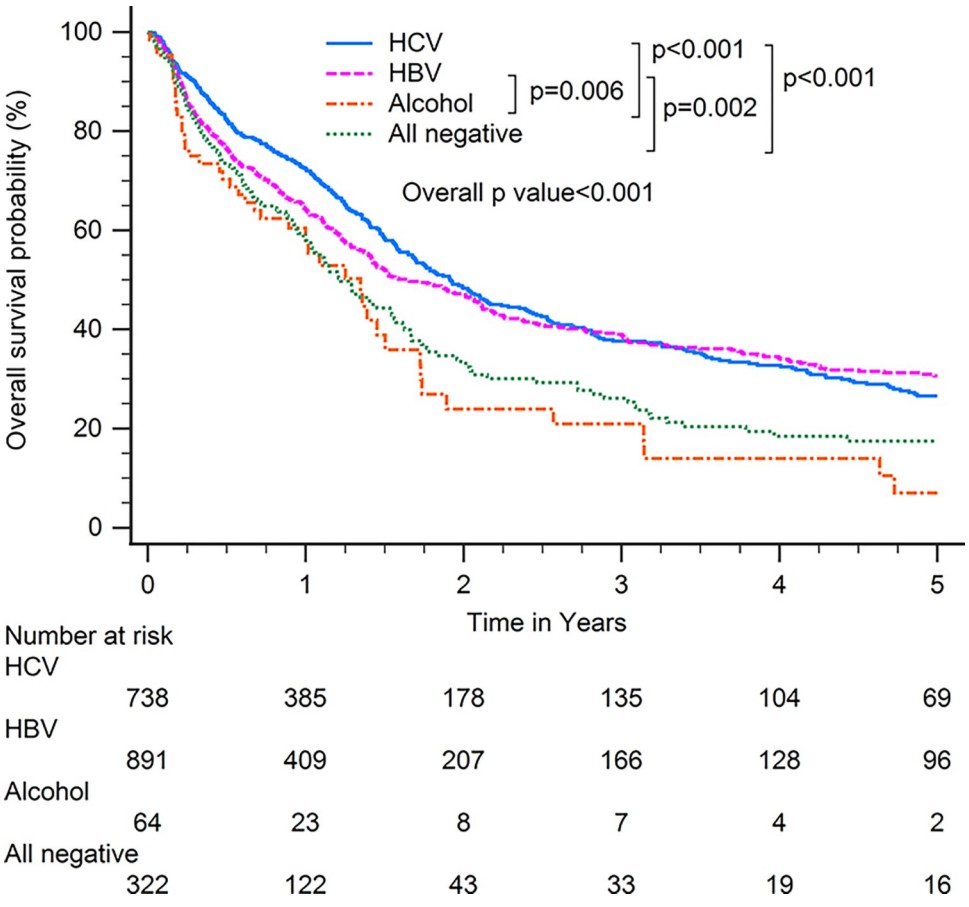

**Fig 3. Overall survival among patients with hepatocellular carcinoma with serum alpha-fetoprotein levels ≧20 ng/ml, according to chronic liver disease etiology.**

observed for the alcohol-related group (HR: 1.029; 95% CI: 0.791–1.339; p<0.830). Multivariate analysis showed that age >65 years (HR: 1.141; 95% CI: 1.030–1.264; p = 0.011), BCLC stages B–D (HR: 2.649; 95% CI: 2.359–2.974; p<0.001), AFP≧20 ng/ml (HR: 1.799; 95% CI: 1.618–1.999; p<0.001), and non-curative treatments (HR: 3.202; 95% CI: 2.869–3.575; p<0.001) remained significant factors for mortality after ad-justing for covariates. Using the all-negative group as the reference group, a significant difference in mortality was observed for the HBV group (HR: 0.856; 95% CI: 0.745–0.983; p = 0.027). No significant differences in mortality were observed for the HCV group (HR: 0.936; 95% CI: 0.814–1.076); p = 0.351) or the alcohol-related group (HR: 1.020; 95% CI: 0.780–1.334; p = 0.885) relative to the all-negative group in the multivariable analysis (Table 2).

## OS in all patients and in subgroups analyses

Patients with HBV-related HCC had superior OS compared with patients with other CLD etiologies when including all patients (p<0.001; Fig 1). In subgroup analysis, patients with HBV-related HCC had superior OS compared with patients with other CLD etiologies among patients with BCLC stages 0–A (p<0.001; Fig 2), serum AFP levels≧20 ng/ml (p<0.001; Fig 3), serum AFP levels <20 ng/ml (p<0.001; Fig 4), age >65 years (p<0.001; Fig 5), and who underwent curative treatments (p = 0.002; Fig 6). No significant differences in OS between patients with HBV-related HCC and those with other HCC etiologies were observed among patients

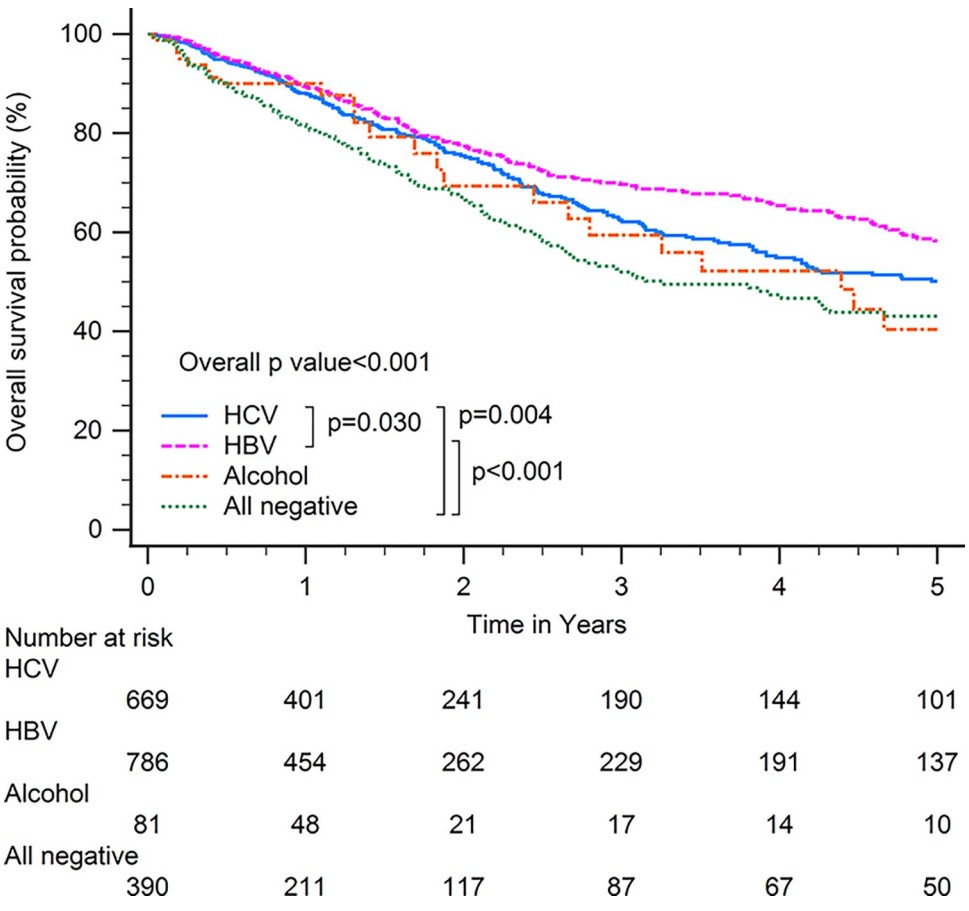

**Fig 4. Overall survival among patients with hepatocellular carcinoma with serum alpha-fetoprotein levels < 20 ng/ml, according to chronic liver disease etiology.**

with age ≤65 years (p = 0.304; Fig 7), BCLC stages B–D (p = 0.973; Fig 8), or who underwent non-curative treatments (p = 0.1; Fig 9).

## Antiviral therapies

Antiviral therapies were administered to 647 patients (46.0%) in the HCV group, which was a significantly smaller proportion than the 1192 (71.0%) patients who were administered nucleos(t)ide analog therapies in the HBV group (p<0.001). Among the 647 patients administered antiviral therapies for HCV, 221 patients (34.2%) received interferon-based therapies only, 333 patients (51.5%) received DAA therapies only, and 93 patients (14.4%) experienced interferon-based therapy failure and were re-treated with DAA therapies.

## Discussion

In the current study, both the multivariate analysis and the majority of subgroup analyses showed that patients with HBV-related HCC had superior OS compared with patients with HCC due to other etiologies. However, no significant differences in OS were observed between patients with HBV-related HCC and patients with other HCC etiologies among subgroups defined by age ≤65 years (p = 0.304), BCLC stages B–D (p = 0.973), or the receipt of non-curative treatments (p = 0.1). Aging is a factor associated with poor prognosis in most chronic

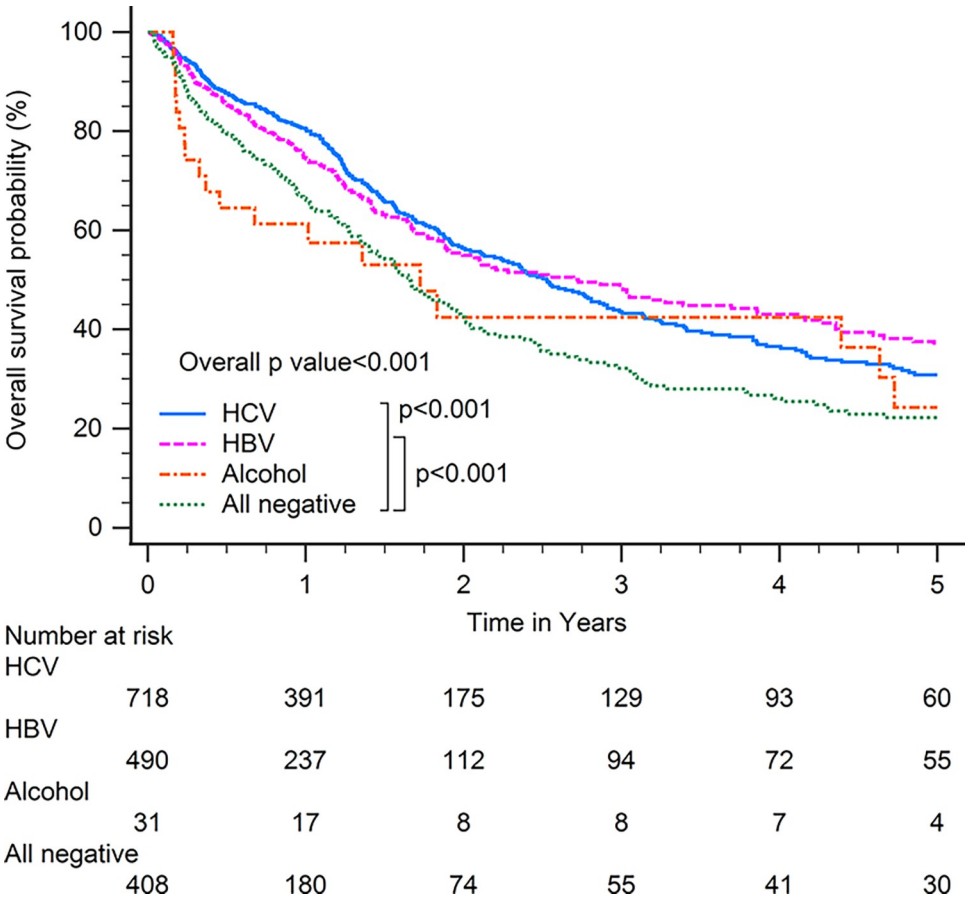

**Fig 5. Overall survival among patients with hepatocellular carcinoma older than 65 years, according to chronic liver disease etiology.**

diseases, including HCC [22]. The association between aging and poor prognosis in HCC patients may be due to the higher risks of more severe comorbidities among older patients. The majority of younger patients (≤65 years) do not present with severe comorbidities irrespective of etiology, with the exception of the alcohol-related group. The largest number of deaths attributable to heavy alcohol intake is associated with cardiovascular disease, followed by injuries, cirrhosis, and cancer [23]. These other causes of death may explain the finding that among patients younger than 65 years, those with alcohol-related HCC had the worst OS among all examined etiologies. In subgroups defined by BCLC stage B–D and non-curative treatments, OS was poor irrespective of etiology due to advanced tumor stage, poor performance status, and poor liver functional reserve.

We speculated that patients with HBV-related HCC may have superior OS compared with patients with other HCC etiologies due to the impacts of antiviral therapy. Antiviral therapy may have potentially beneficial effects following the application of curative treatments for HBV-related HCC, resulting in improvements in recurrence-free survival, OS, and liver-related mortality [24]. The majority of patients with HBV-related HCC (71.0%) in our study received antiviral therapies. By contrast, HCV-related HCC was not identified as an independent factor associated with mortality in the present study. In our subgroups analyses, patients with HCV-related HCC had inferior OS compared with patients with HBV-related HCC among patients with BCLC stages 0–A (p<0.001), AFP < 20 ng/ml (p = 0.03), and curative

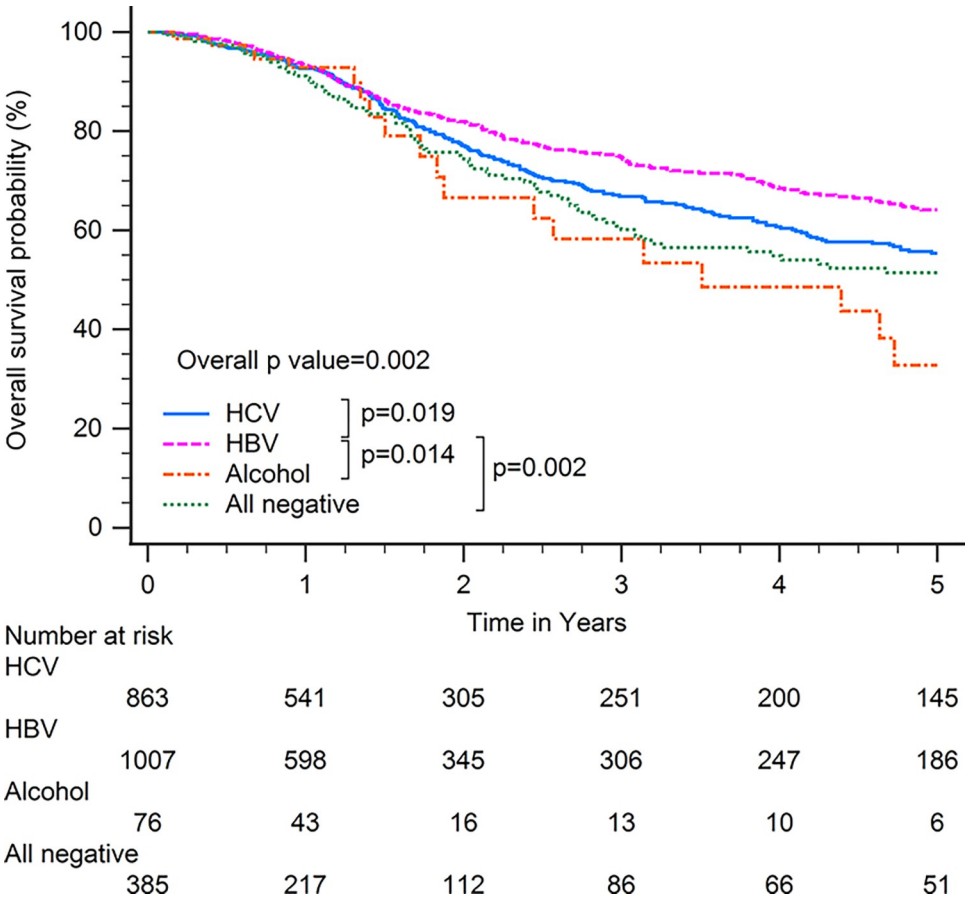

**Fig 6. Overall survival among patients with hepatocellular carcinoma who underwent curative treatments, according to chronic liver disease etiology.**

treatments (p = 0.019). Although interferon-based therapies improve recurrence-free survival among HCC patients who undergo resection or ablation [25], in our clinical experience, patients with HCV-related HCC are often considered too frail to receive interferon-based therapies. Only 314 (22.3%) patients with HCV-related HCC in our cohort received interferon-based therapies. The government began to reimburse the costs of DAA therapy starting in January 2017. An earlier study suggested a potential increased risk of HCC recurrence after DAA treatment in patients with HCV infection and prior history of treated HCC who achieved a complete response [26]. However, a review did not highlight a higher rate of HCC recurrence after DAA therapy in patients with previous HCV infection [27].

Compared with the groups associated with other CLD etiologies, the all-negative group was the oldest and presented with the largest tumor size, the smallest proportion of early-stage HCC (i.e., BCLC stages 0 and A), the highest BMI, and the largest proportion of non-cirrhotic livers, which is consistent with the characteristics of NAFLD-related HCC [12, 13, 28]. The most common non-viral HCC etiologies are alcohol-related and NAFLD [21]. Therefore, we expect that the majority of patients in the all-negative group have NAFLD-related HCC. A recent Italian study demonstrated the changing scenario of HCC etiology in the last 15 years characterized by a progressive increase in non-viral cases, in particular, "metabolic" and "metabolic + alcohol" HCC cases, and progressive patient ageing [29]. Similar results have also

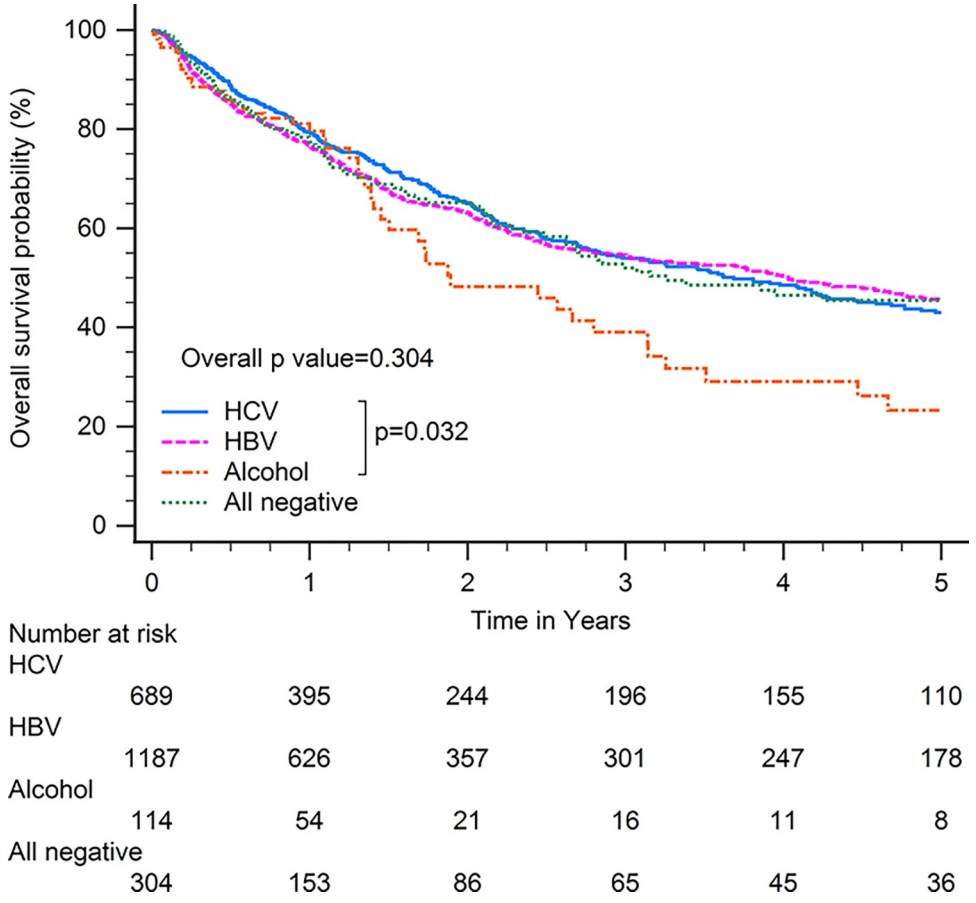

**Fig 7. Overall survival among patients with hepatocellular carcinoma 65 years or younger, according to chronic liver disease etiology.**

been noted in Taiwan; the proportion of patients with non-HBV- and non-HCV (NBNC)-HCC increased in the period from 2011 to 2020 at our institution [30].

Whether CLD etiologies impact OS in patients with HCC remains controversial. Hester et al. found no significant difference in OS between patients with NAFLD-related HCC and other etiologies after multivariate analysis [28]. Karim et al. reported that patients with NAFLD-related HCC were associated with worse OS than patients with HCV-related HCC (adjusted HR: 1.20; 95% CI: 1.09–1.32) [13]. Benhammou et al. demonstrated that patients with NAFLD-related HCC had superior OS compared with both patients with HCV-related HCC (adjusted HR: 0.37, 95% CI: 0.17–0.77, p = 0.003) and patients with HBV-related HCC (adjusted HR: 0.35, 95% CI: 0.15–0.80, p = 0.013) [31]. Piscaglia et al. conducted a multicenter prospective study, which showed no significant difference in OS between patients with HCV-related HCC and those with NAFLD-related HCC after propensity score matching [32]. In the current study, patients with HBV-related HCC had superior OS compared with the all-negative group (adjusted HR: 0.856; 95% CI: 0.745–0.983; p = 0.027).

Previous studies [6, 8, 9] reported that viral-related HCC (especially HBV-related HCC) might display more aggressive tumor behavior, resulting in a theoretically worse prognosis compared with other etiologies. However, we found the opposite results in the present study. One possible explanation is that hepatic inflammation is a driving force for hepatocarcinogenesis [33]. Antiviral therapies have also been shown to improve the prognosis of patients with

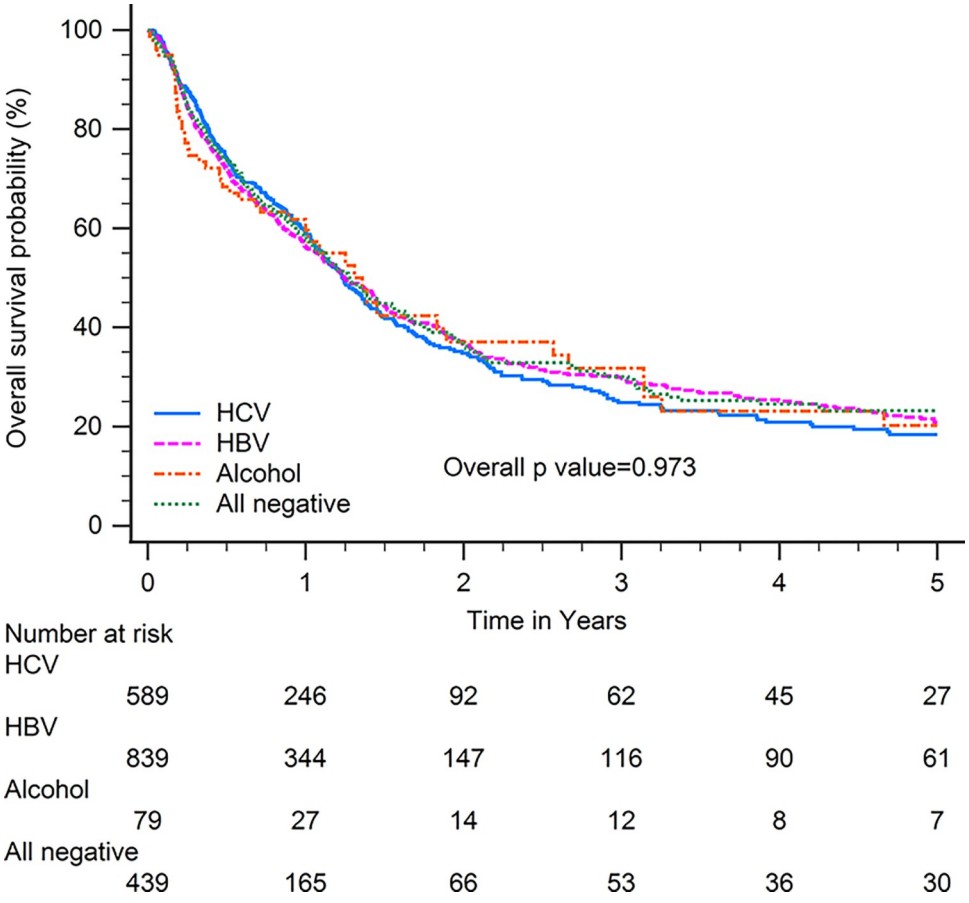

**Fig 8. Overall survival among patients with hepatocellular carcinoma classified as Barcelona clinic liver stages B–D, according to chronic liver disease etiology.**

virus-related HCC who underwent curative treatments in prior studies [24, 25, 34]. Currently, no effective treatments have been developed for NAFLD- or alcohol-related CLD [23, 35]. Weight loss and abstinence from alcohol use are the most effective approaches for treating NAFLD- and alcohol-related CLD, respectively [23, 35], which can be difficult for many patients. The availability of additional treatments could also explain why the virus-related HCC group had superior OS compared with the alcohol-related and all-negative groups in the current study.

The degree of liver dysfunction is an independent prognostic factor of patients with HCC. A detailed assessment should be performed for patients with worse liver functional reserve. Such an assessment should balance the expected antitumor efficacy of therapy and the risk of deterioration in liver functional reserve [36, 37].

One strength of the present study is the use of a large cohort of patients with HCC that is associated with prospectively collected data and limited missing data. However, the study also has several limitations. First, the study lacked a complete list of comor-bidities for the included patients. Second, we did not have data on hepatic steatosis, metabolic risk factors (e.g., hypertension, dyslipidemia, central obesity, and hypergly-cemia) [33], or possible CLD etiologies other than HBV, HCV, and alcohol-related. Therefore, we were unable to define NAFLD in the present study. Third, for patients with multiple CLD etiologies, the primary etiology was classified using the following hierarchy: HCV > HBV > alcohol-related > all negative.

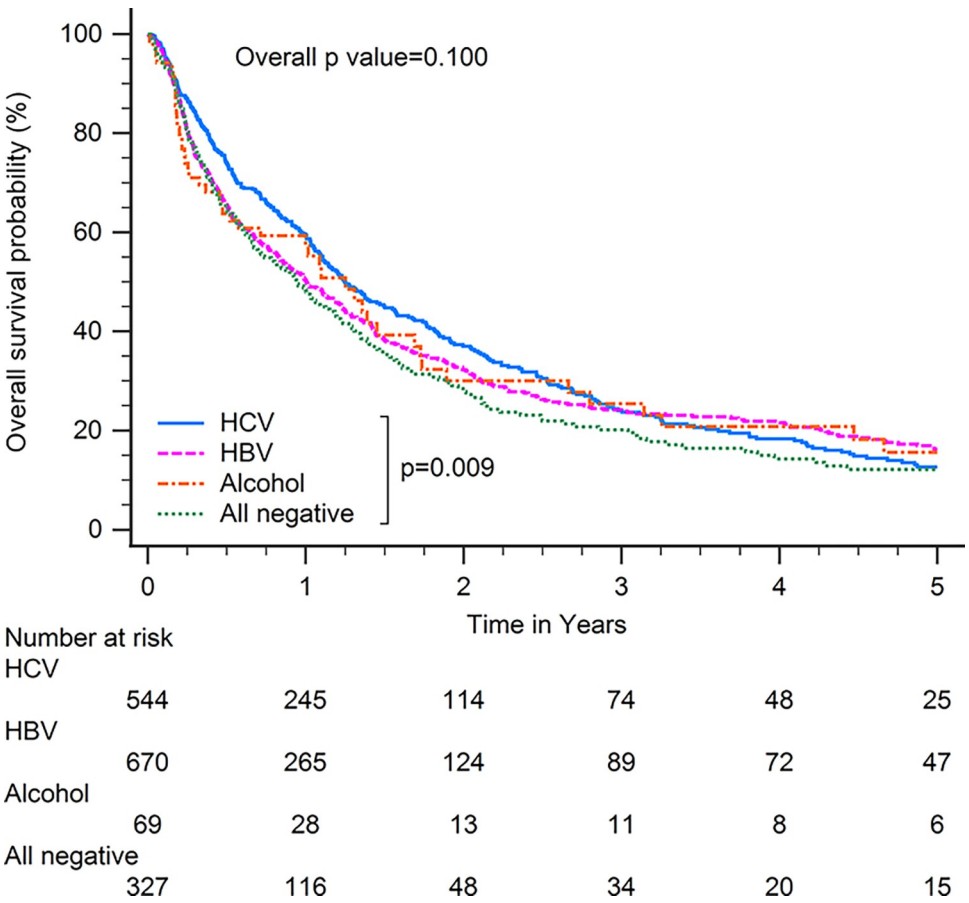

**Fig 9. Overall survival among patients with hepatocellular carcinoma who underwent non-curative treatments, according to chronic liver disease etiology.**

Although this method simplified etiology classification, 176 patients in the HCV group were co-infected with both HBV and HCV. In addition, of the 1407 patients classified with HCV-related HCC, 125 (8.9%) had alcohol use disorder; of the 1677 patients who had HBV-related HCC, 216 (12.9%) also had alcohol use disorder. A previous study showed that ethanol intake is an independent predictor of liver cirrhosis in subjects with chronic HCV infection and an independent predictor of death of subjects with either HCV or HBV infection [38]. Therefore, the prognosis of patients with underlying viral (HBV, HCV)-related HCC with alcohol use disorder as an adjunctive etiology might be worse than that of patients with viral (HBV, HCV)-related HCC but without alcohol use disorder. Fourth, the study lacked data on HCV RNA. Some proportion of patients who were positive for anti-HCV antibodies may represent patients who have experienced a prior episode of HCV infection that has resolved rather than patients who are experiencing an active infection. Fifth, this study was conducted as a retrospective and monocentric study. Sixth, we did not record the amount of alcohol intake, and the case numbers in the alcohol-related group were limited, which may result in the underestimation of the impacts of alcohol use on CLD. In addition, we could not ascertain whether the patients were active consumers of alcohol or whether alcohol consumption was properly managed because this information is not recorded in our HCC registry dataset. These behaviors might be associated with the outcomes of these patients. Seventh, the present results are mostly derived from Eastern patients. HBV infection is more common in Eastern than Western

countries, where the hepatitis B mass vaccination has reduced the number of patients with HBV-related cirrhosis. This difference could make this study's data less appreciated in the United States and Europe. Finally, while multivariate analysis as used to evaluate whether CLD's etiology impacts the OS of patients with HCC, the HBV group was younger than the HCV group and all negative groups. Furthermore, compared to the groups associated with other CLD etiologies, patients in the "all-negative"group were the oldest and presented with the largest tumor sizes and smallest proportion of early-stage HCC. Therefore, this study's results should be interpreted cautiously.

## Conclusion

In this large-scale, contemporary study, we found that patients with HBV-related HCC had superior OS compared with patients with other CLD etiologies. The results of the current study may be due to the widespread use of nucleos(t)ide an-alog therapies in patients with HBV-related HCC.

## Supporting information

**S1 Data.**
(XLSX)

## Acknowledgments

The authors thank Cancer Center, Kaohsiung Chang Gung Memorial Hospital for the provision of HCC registry data. The authors thank Chih-Yun Lin and Nien-Tzu Hsu and the Biostatistics Center, Kaohsiung Chang Gung Memorial Hospital for statistics work.

## Author Contributions

**Conceptualization:** Yi-Hao Yen.

**Formal analysis:** Chih-Yun Lin.

**Supervision:** Kwong-Ming Kee, Tsung-Hui Hu, Ming-Chao Tsai, Yuan-Hung Kuo, Wei-Feng Li, Yueh-Wei Liu, Chih-Chi Wang.

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
