## [Decision Letter · Decision Letter 0]

15 Nov 2023

PONE-D-23-24865Hepatitis B virus–related hepatocellular carcinoma has superior overall survival compared with other etiologiesPLOS ONE

Dear Dr. Yen,

Thank you for submitting your manuscript to PLOS ONE. After careful consideration, we feel that it has merit but does not fully meet PLOS ONE’s publication criteria as it currently stands. Therefore, we invite you to submit a revised version of the manuscript that addresses the points raised during the review process.

We look forward to receiving your revised manuscript.

Kind regards,

Alessandro Granito

Academic Editor

PLOS ONE

https://www.mdpi.com/2072-6694/15/6/1687/html

In your revision ensure you cite all your sources (including your own works), and quote or rephrase any duplicated text outside the methods section. Further consideration is dependent on these concerns being addressed.

Reviewers' comments:

Reviewer's Responses to Questions

**Comments to the Author**

1. Is the manuscript technically sound, and do the data support the conclusions?

Reviewer #1: Partly

Reviewer #2: Yes

2. Has the statistical analysis been performed appropriately and rigorously? 

Reviewer #1: Yes

Reviewer #2: Yes

3. Have the authors made all data underlying the findings in their manuscript fully available?

Reviewer #1: Yes

Reviewer #2: Yes

4. Is the manuscript presented in an intelligible fashion and written in standard English?

Reviewer #1: Yes

Reviewer #2: Yes

5. Review Comments to the Author

Reviewer #1: In this study, the authors aimed to assess whether the etiology of chronic liver disease (CLD) impacts the overall

survival (OS) of patients with hepatocellular carcinoma (HCC). Overall, 3941 HCC patients who were newly

diagnosed with HCC were enrolled. The etiology of lived diseases was classified as hepatitis C virus (HCV), hepatitis B virus (HBV), alcohol-related and as all negative (negative for HCV, HBV, and alcohol use disorder). They found that among 3941 patients, 1407 patients were classified with HCV-related HCC, 1677 patients had HBV-related HCC, 145 patients had alcohol-related HCC, and 712 patients had all-negative HCC. Patients with HBV-related HCC showed significantly superior OS compared with patients with other CLD etiologies. Subgroup analyses were performed, for Barcelona Clinic Liver Cancer (BCLC) stages 0–A (p<0.001); serum alpha-fetoprotein (AFP) levels≧20 ng/ml

(p<0.001); AFP levels < 20 ng/ml (p<0.001); age > 65 years (p<0.001); and the use of curative treatments (p=0.002). No significant difference in OS between HBV and other etiologies was observed among patients aged ≤ 65 years (p=0.304); with BCLC stages B–D (p=0.973); or who underwent non-curative treatments (p=0.1).

They concluded that patients with HBV-related HCC had superior OS than patients with other HCC etiologies.

The study is of interest and include a very large study population. However, several issues need further informations and should be addressed.

1) Patients categories: 4 distinct etiological categories were considered. However, it si well-known that simultaneous etiologies may concur to liver disease prognosis in the same patient. In particular, alcohol intake is not rarely reported as adjunctive etiology in patients with underlying viral (HBV, HCV) chronic liver diseases. The authors should therefore clearly describe whether alcohol intake was evaluated and excluded in all patient group. Moreover, discussing their clinically relevant results, the authors should highlight the important role of alcohol intake in patients with viral liver disease and recall a previous pivotal study demonstrating that ethanol intake is an independent predictor of liver cirrhosis in subjects with chronic HCV infection and an independent predictor of death in subjects with either HCV or HBV infection, as previously demonstrated (Natural course of chronic HCV and HBV infection and role of alcohol in the general population: the Dionysos Study. Am J Gastroenterol. 2008 Sep;103(9):2248-53.).

-Regarding the alcohol-related group it would be also relevant to report whether the authors could ascertain if alcohol consumption was still active or patients properly managed.

-HCV and HBV patients: the authors stated that antiviral therapies were administered to 647 patients (46.0%) in the HCV group, which was a significantly smaller proportion than the 1192 (71.0%) patients who were administered nucleos(t)ide analog therapies in the HBV group (p<0.001). A potential explanation of the better prognosis of HBV patients might be due to the efficacy of antiviral treatment. Taking into account the subgroup of HCV and HBV underwent antiviral treatments, there were significant differences in OS between HBV and HCV patients according to the residual liver function (Child-Pugh class)?

-Body max index: this is a clinically relevant point of the study. As Body mass index (BMI) was higher in the all-negative group (p=0.032) than in the other groups, it could be speculated that "all negative (negative for HCV, HBV, and alcohol use disorder)" patient group had an underlying NASH-related chronic liver disease?. In this regard, the author should recall recent literature data demonstrating the changing scenario of HCC etiology in the last 15 years characterized by (a a progressive increase of non-viral cases and, particularly, of "metabolic" and "metabolic + alcohol" HCCs and a progressive patient ageing, as recently demonstrated (The changing scenario of hepatocellular carcinoma in Italy: an update. Liver Int. 2021 Mar;41(3):585-597. doi: 10.1111/liv.14735.).

-the last point worth mentioning and discussing is the impact of liver functional reserve in the non-surgical treatment of hepatocellular carcinoma. Discussing their findings, the authors should recall the role of functional assessment for each type of therapy for HCC as recently described (The importance of liver functional reserve in the non-surgical treatment of hepatocellular carcinoma. J Hepatol. 2022 May;76(5):1185-1198.), in particular in patients with more deteriorated liver function such as those with Child-Pugh Class B (Non-transplant therapies for patients with hepatocellular carcinoma and Child-Pugh-Turcotte class B cirrhosis. Lancet Oncol. 2017 Feb;18(2):e101-e112. ).

Reviewer #2: Thank you for the opportunity to review the manuscript Hepatitis B virus–related hepatocellular carcinoma has superior overall survival compared with other etiologies.

The subject matter is interesting and important. The paper is well written. The sections are readable and meet the requirements.

Please find my comments below:

Please state why you wish to clarify the research questions? Please state the implications of your study? Please discuss how the results are important.

What was the dose of the antiviral drug?

6. PLOS authors have the option to publish the peer review history of their article (what does this mean?). If published, this will include your full peer review and any attached files.

Reviewer #1: No

Reviewer #2: No

---

## [Author Response · Author response to Decision Letter 0]

1 Dec 2023

Reviewer #1: In this study, the authors aimed to assess whether the etiology of chronic liver disease (CLD) impacts the overall

survival (OS) of patients with hepatocellular carcinoma (HCC). Overall, 3941 HCC patients who were newly

diagnosed with HCC were enrolled. The etiology of lived diseases was classified as hepatitis C virus (HCV), hepatitis B virus (HBV), alcohol-related and as all negative (negative for HCV, HBV, and alcohol use disorder). They found that among 3941 patients, 1407 patients were classified with HCV-related HCC, 1677 patients had HBV-related HCC, 145 patients had alcohol-related HCC, and 712 patients had all-negative HCC. Patients with HBV-related HCC showed significantly superior OS compared with patients with other CLD etiologies. Subgroup analyses were performed, for Barcelona Clinic Liver Cancer (BCLC) stages 0–A (p<0.001); serum alpha-fetoprotein (AFP) levels≧20 ng/ml

(p<0.001); AFP levels < 20 ng/ml (p<0.001); age > 65 years (p<0.001); and the use of curative treatments (p=0.002). No significant difference in OS between HBV and other etiologies was observed among patients aged ≤ 65 years (p=0.304); with BCLC stages B–D (p=0.973); or who underwent non-curative treatments (p=0.1).

They concluded that patients with HBV-related HCC had superior OS than patients with other HCC etiologies.

The study is of interest and include a very large study population. However, several issues need further informations and should be addressed.

1) Patients categories: 4 distinct etiological categories were considered. However, it si well-known that simultaneous etiologies may concur to liver disease prognosis in the same patient. In particular, alcohol intake is not rarely reported as adjunctive etiology in patients with underlying viral (HBV, HCV) chronic liver diseases. The authors should therefore clearly describe whether alcohol intake was evaluated and excluded in all patient group. Moreover, discussing their clinically relevant results, the authors should highlight the important role of alcohol intake in patients with viral liver disease and recall a previous pivotal study demonstrating that ethanol intake is an independent predictor of liver cirrhosis in subjects with chronic HCV infection and an independent predictor of death in subjects with either HCV or HBV infection, as previously demonstrated 

Response: Thank you for your comments. 

Of the 1407 patients classified with HCV-related HCC, 125 (8.9%) had alcohol use disorder; of the 1677 patients who had HBV-related HCC, 216 (12.9%) also had alcohol use disorder. A previous study showed that ethanol intake is an independent predictor of liver cirrhosis in subjects with chronic HCV infection and an independent predictor of death of subjects with either HCV or HBV infection [38]. Therefore, the prognosis of patients with underlying viral (HBV, HCV)-related HCC with alcohol use disorder as an adjunctive etiology might be worse than that of patients with viral (HBV, HCV)-related HCC but without alcohol use disorder. Please see page 26. 

-Regarding the alcohol-related group it would be also relevant to report whether the authors could ascertain if alcohol consumption was still active or patients properly managed.

Response: We could not ascertain whether the patients were active consumers of alcohol or whether alcohol consumption was properly managed because this information is not recorded in our HCC registry dataset. These behaviors might be associated with the outcomes of these patients. Please see page 27. 

-HCV and HBV patients: the authors stated that antiviral therapies were administered to 647 patients (46.0%) in the HCV group, which was a significantly smaller proportion than the 1192 (71.0%) patients who were administered nucleos(t)ide analog therapies in the HBV group (p<0.001). A potential explanation of the better prognosis of HBV patients might be due to the efficacy of antiviral treatment. Taking into account the subgroup of HCV and HBV underwent antiviral treatments, there were significant differences in OS between HBV and HCV patients according to the residual liver function (Child-Pugh class)?

Response: In the subgroup of patients who underwent antiviral treatments, the 5-year OS of HCV patients was superior to that of HBV patients among those with Child–Pugh class A liver disease (70% vs 48%, p<0.001); in addition, the 5-year OS of HCV patients was superior to that of HBV patients among those with Child–Pugh class B or C liver disease (44% vs 16%, p<0.001). For further information on this subject, see below. 

Previous studies have suggested that patients receiving antiviral therapy have a reduced risk of recurrence and mortality compared to untreated patients. [24, 25] Theoretically, the overall survival of the subgroup of patients undergoing antiviral treatments is associated with tumor burden, treatments received, performance status, etc. 

References 

24. Wong JS, Wong GL, Tsoi KK, Wong VW, Cheung SY, Chong CN, et al. Meta-analysis: the efficacy of anti-viral therapy in prevention of recurrence after curative treatment of chronic hepatitis B-related hepatocellular carcinoma. Aliment Pharmacol Ther 2011; 33: 1104–1112. doi: 10.1111/j.1365-2036.2011.04634.x

25. Shen YC, Hsu C, Chen LT, Cheng CC, Hu FC, Cheng AL. Adjuvant interferon therapy after curative therapy for hepatocellular carcinoma (HCC): A meta-regression approach. J Hepatol 2010; 52: 889–894. doi: 10.1016/j.jhep.2009.12.041.

Figure 1. In the subgroup of patients who underwent antiviral treatments, the 5-year OS of HCV patients was superior to that of HBV patients among those with Child–Pugh class A liver disease (70% vs 48%, p<0.001).

Figure 2. In the subgroup of patients who underwent antiviral treatments, the 5-year OS of HCV patients was superior to that of patients with HBV among those with Child–Pugh class B or C liver disease (44% vs 16%, p<0.001).

-Body max index: this is a clinically relevant point of the study. As Body mass index (BMI) was higher in the all-negative group (p=0.032) than in the other groups, it could be speculated that "all negative (negative for HCV, HBV, and alcohol use disorder)" patient group had an underlying NASH-related chronic liver disease?. In this regard, the author should recall recent literature data demonstrating the changing scenario of HCC etiology in the last 15 years characterized by (a a progressive increase of non-viral cases and, particularly, of "metabolic" and "metabolic + alcohol" HCCs and a progressive patient ageing, as recently demonstrated (The changing scenario of hepatocellular carcinoma in Italy: an update. Liver Int. 2021 Mar;41(3):585-597. doi: 10.1111/liv.14735.).

Response: 

A recent Italian study demonstrated the changing scenario of HCC etiology in the last 15 years characterized by a progressive increase in non-viral cases, in particular, “metabolic” and “metabolic + alcohol” HCC cases, and progressive patient ageing [29]. Similar results have also been noted in Taiwan; the proportion of patients with non-HBV- and non-HCV (NBNC)-HCC increased in the period from 2011 to 2020 at our institution [30]. Please see page 23, last paragraph. 

-the last point worth mentioning and discussing is the impact of liver functional reserve in the non-surgical treatment of hepatocellular carcinoma. Discussing their findings, the authors should recall the role of functional assessment for each type of therapy for HCC as recently described (The importance of liver functional reserve in the non-surgical treatment of hepatocellular carcinoma. J Hepatol. 2022 May;76(5):1185-1198.), in particular in patients with more deteriorated liver function such as those with Child-Pugh Class B (Non-transplant therapies for patients with hepatocellular carcinoma and Child-Pugh-Turcotte class B cirrhosis. Lancet Oncol. 2017 Feb;18(2):e101-e112. ).

Response: 

The degree of liver dysfunction is an independent prognostic factor of patients with HCC. A detailed assessment should be performed for patients with worse liver functional reserve. Such an assessment should balance the expected antitumor efficacy of therapy and the risk of deterioration in liver functional reserve [36, 37]. Please see page 25, 2nd paragraph. 

Reviewer #2: Thank you for the opportunity to review the manuscript Hepatitis B virus–related hepatocellular carcinoma has superior overall survival compared with other etiologies.

The subject matter is interesting and important. The paper is well written. The sections are readable and meet the requirements.

Response: We are grateful for your comments.

---

## [Decision Letter · Decision Letter 1]

18 Dec 2023

Hepatitis B virus–related hepatocellular carcinoma has superior overall survival compared with other etiologies

PONE-D-23-24865R1

Dear Dr. Yen,

We’re pleased to inform you that your manuscript has been judged scientifically suitable for publication and will be formally accepted for publication once it meets all outstanding technical requirements.

Kind regards,

Alessandro Granito

Academic Editor

PLOS ONE

Additional Editor Comments (optional):

Reviewers' comments:

Reviewer's Responses to Questions

**Comments to the Author**

1. If the authors have adequately addressed your comments raised in a previous round of review and you feel that this manuscript is now acceptable for publication, you may indicate that here to bypass the “Comments to the Author” section, enter your conflict of interest statement in the “Confidential to Editor” section, and submit your "Accept" recommendation.

Reviewer #1: All comments have been addressed

Reviewer #2: All comments have been addressed

2. Is the manuscript technically sound, and do the data support the conclusions?

Reviewer #1: Yes

Reviewer #2: Yes

3. Has the statistical analysis been performed appropriately and rigorously? 

Reviewer #1: Yes

Reviewer #2: Yes

4. Have the authors made all data underlying the findings in their manuscript fully available?

Reviewer #1: Yes

Reviewer #2: Yes

5. Is the manuscript presented in an intelligible fashion and written in standard English?

Reviewer #1: Yes

Reviewer #2: Yes

6. Review Comments to the Author

Reviewer #1: In this revised version, The authors have properly addressed the raised points and the manuscript can be accepted.

Reviewer #2: Authors address an important topic of interest to a wide audience. The manuscript is well written and the conclusions are supported by the data. Statistical analysis is carried out appropriately.

7. PLOS authors have the option to publish the peer review history of their article (what does this mean?). If published, this will include your full peer review and any attached files.

Reviewer #1: No

Reviewer #2: No

---

## [Editor Report · Acceptance letter]

28 Dec 2023

PONE-D-23-24865R1 

PLOS ONE

Dear Dr. Yen, 

I'm pleased to inform you that your manuscript has been deemed suitable for publication in PLOS ONE. Congratulations! Your manuscript is now being handed over to our production team.

Kind regards, 

on behalf of

Dr. Alessandro Granito 

Academic Editor

PLOS ONE